# Biochemical Characterization of Orange-Colored Rice Calli Induced by Target Mutagenesis of *OsOr* Gene

**DOI:** 10.3390/plants12010056

**Published:** 2022-12-22

**Authors:** Hee Kyoung Kim, Jin Young Kim, Jong Hee Kim, Ji Yun Go, Yoo-Seob Jung, Hyo Ju Lee, Mi-Jeong Ahn, Jihyeon Yu, Sangsu Bae, Ho Soo Kim, Sang-Soo Kwak, Me-Sun Kim, Yong-Gu Cho, Yu Jin Jung, Kwon Kyoo Kang

**Affiliations:** 1Division of Horticultural Biotechnology, Hankyong National University, Anseong 17579, Republic of Korea; 2College of Pharmacy and Research Institute of Pharmaceutical Sciences, Gyeongsang National University, Jinju 52828, Republic of Korea; 3Division of Life Sciences, Korea Polar Research Institute, Incheon 21990, Republic of Korea; 4Department of Biochemistry and Molecular Biology, Department of Biomedical Sciences, Seoul National University College of Medicine, Seoul 03080, Republic of Korea; 5Plant Systems Engineering Research Center, Korea Research Institute of Bioscience and Biotechnology (KRIBB), 125 Gwahak-ro, Yuseong-gu, Daejeon 34141, Republic of Korea; 6Department of Crop Science, Chungbuk National University, Cheongju 28644, Republic of Korea; 7Institute of Genetic Engineering, Hankyong National University, Anseong 17579, Republic of Korea

**Keywords:** CRISPR-Cas9, carotenoids, orange gene (*OsOr*), orange-colored rice calli

## Abstract

We generated an orange-colored (OC) rice callus line by targeted mutagenesis of the orange gene (*OsOr*) using the CRISPR-Cas9 system. The OC line accumulated more lutein, *β*-carotene, and two *β*-carotene isomers compared to the WT callus line. We also analyzed the expression levels of carotenoid biosynthesis genes by qRT-PCR. Among the genes encoding carotenoid metabolic pathway enzymes, the number of transcripts of the *PSY2*, *PSY3*, *PDS*, *ZDS* and *β-LCY* genes were higher in the OC line than in the WT line. In contrast, transcription of the *ε-LCY* gene was downregulated in the OC line compared to the WT line. In addition, we detected increases in the transcript levels of two genes involved in carotenoid oxidation in the OC lines. The developed OC lines also showed increased tolerance to salt stress. Collectively, these findings indicate that targeted mutagenesis of the *OsOr* gene via CRISPR/Cas9-mediated genome editing results in *β*-carotene accumulation in rice calli. Accordingly, we believe that this type of genome-editing technology could represent an effective alternative approach for enhancing the *β*-carotene content of plants.

## 1. Introduction

In plants, carotenoids play important roles in photosynthesis as auxiliary pigments that harvest energy from the sun and subsequently deliver it to chlorophyll [1,2]. In addition, these pigments not only act as precursors in abscisic acid (ABA) biosynthesis but are also a rich source of provitamin A and thus are of nutritional importance in human diets [3,4]. Carotenoids also play a protective role by interacting with reactive oxygen species to prevent oxidative damage to cells, and serve as useful ecological signals that can be used by pollination and seed dispersal agents, such as insects and birds, to distinguish different plants [5]. These pigments tend to accumulate in specific organs such as fruits, roots, and flowers, thereby conferring their characteristic coloration [6,7,8,9,10]. Carotenoids, which accumulate as red, orange, and yellow pigments, are synthesized in plastids, among which *β*-carotene, lycopene, and lutein are important constituents exploited by the food and petroleum industries owing to their powerful antioxidant activities [11].

Rice is a staple food crop and a major source of nutrition for more than 40% of the world’s population. Numerous studies have been conducted to examine the accumulation of carotenoids in the endosperm rice grains [8,12]. Although it has been established that rice endosperm is not normally involved in carotenoid synthesis, it has been reported that not only geranylgeranyl diphosphate (GGPP), a carotenoid precursor, but also several genes in the carotenoid biosynthetic pathway, including *β-LCY*, are found in rice endosperm [8,13]. For efficient biosynthesis of *β*-carotene in rice endosperm, transgenic Golden Rice 2 was developed, which expresses the maize phytoene synthase (*PSY*) and bacterial *CrtI* genes under the regulation of the glutelin promoter [8]. Enhanced levels of zeaxanthin, *β*-carotene, and total carotenoid contents have also been achieved in transgenic potatoes by controlling the expression of *LCY-b*, *CHY-ε*, and *ZEP* [14,15]. In addition, tomato plants overexpressing the *PSY*, *LCY*-*b*, and *CHY*-*b* genes were characterized by dwarf phenotypes and fruits with higher contents of lycopene, carotene, and zeaxanthin than the control plants [16]. The orange (*Or*) gene of plants was first isolated from an orange cauliflower mutant that showed variation in the content of *β*-carotene [17]. Since this discovery, the gene has been identified in numerous plant species and has been reported to encode cysteine-rich DnaJ proteins confined to plastids and regulate carotenoid accumulation and abiotic stress resistance in various plant species [18]. Recently, overexpression of the *Arabidopsis thaliana Or* gene in conjunction with that of the *PSY* and *crtl* genes was found to enhance carotenoid accumulation in rice calli and maize seeds [13,19]. A single amino acid alteration in wild-type OR significantly enhanced carotenoid accumulation in potato tubes, sweet potato calli, and the melon *Cucumis melo* [5,17,18,20]. *AtOr-His*, a variant of the gene that carries golden SNPs in *Arabidopsis thaliana*, was overexpressed in *Arabidopsis* callus and tomato (*Solanum lycopersicum*), leading to a high-level accumulation of carotenoids [21,22]. In addition to its role in carotenoid accumulation, orange gene expression in plants enhances tolerance to environmental stress. For example, in sweet potatoes, overexpression of *IbOr* has been found to enhance tolerance to heat and oxidative stress [23,24], whereas in alfalfa and potato, overexpression of *IbOr* has been observed to confer resistance to abiotic stress factors, such as salt, drought, and heat [25,26]. Recently, Endo et al. successfully increased *β*-carotene accumulation in rice calli by directly modifying the rice *OsOr* gene via CRISPR/Cas9-mediated genome editing, which disrupted the proper splicing of *OsOr*, similarly to the cauliflower [17]. In addition, Yu et al. reported that overexpression of the rice orange gene *OsOr* negatively regulated carotenoid accumulation, leading to higher tiller numbers and decreased stress tolerance in Nipponbare rice [27].

In this study, we generated an orange-colored (OC) rice callus line by targeted mutagenesis of the *OsOr* gene using the CRISPR/Cas9 system. Our results showed that OC lines contained elevated total carotenoid content, with significant increases in the levels of lycopene, *β*-carotene, and the two *β*-carotene isomers. Moreover, we performed gene expression analysis using quantitative RT-PCR to understand the mechanisms associated with the accumulation of orange color.

## 2. Results

### 2.1. CRISPR/Cas9 Targeted Mutagenesis of OsOr

To understand the role of *OsOr* in the mechanisms associated with the accumulation of orange color, we created an orange-colored (OC) rice callus line by targeted mutagenesis of the *OsOr* using the CRISPR/Cas9 system. *OsOr* encodes a DnaJ cysteine-rich zinc-binding domain consisting of eight exons (Figure 1A). To obtain *OsOr* mutants, three single guide RNAs (sgRNAs), sgRNA1, sgRNA2, and sgRNA3, were designed to target the first and second exons, and a region between the third exon and the third intron, respectively (Figure 1A). Three pBOsC binary vectors containing each sgRNA were used for transformation of the rice cultivar Dongjin, as reported previously by Jung et al. [28] (Figure 1B). We generated 30 T_0_ transformants for constructs containing sgRNA1, sgRNA2, and sgRNA3 and identified the integration of T-DNA into genomic DNA using PCR-based genotyping with *bar* gene-specific primers (Appendix A). Among them, the orange color phenotype was observed in the transgenic callus obtained from the sgRNA3 construct (Figure 1C).

Next, we performed a deep-sequencing analysis of all transgenic calluses to analyze target-site mutations. The mutation rates of sgRNA1, sgRNA2, and sgRNA3 were 36.8%, 40.1%, and 43.5%, respectively (Appendix A). Deep sequencing results of the target region showed various mutations involving insertions (1–2 bp) and deletions (1–65 bp) of different nucleotides (Figure 2). The most common types of mutations were 1 bp insertions and deletions (indel) (Figure 2). These mutations were classified as homologous, heterologous, and biallelically edited (Figure 2). We also investigated sgRNAs using Cas-OFFinder (https://www.rgenome.net/cas-offinder/, accessed on 26 November 2019) [28] and selected two potential off-target sites. However, we were unable to identify off-target mutations after sequencing of these loci (data not shown). 

Among sgRNA3-derived transgenic calli, three independent orange calli (−65,*1/−65,*1; −55/−55; −41/−41) with large deletions were selected. For further study, these calli were subcultured several times and finally named OC#1, OC#3, and OC#17 (Figure 3A, Appendix A). In addition, we extracted pigments from the WT, OC#1, OC#3, and OC#17 lines using acetone and measured the absorption spectra of each extract (Figure 3B). The extracts from WT showed weak absorption, with λmax at 450 nm. The extracts from the OC#1, OC#3, and OC#17 lines showed higher levels of absorption than the WT extract, with λmax at 425, 452, and 475 nm, respectively. The three callus lines showed similar orange absorption spectra (Figure 3B).

### 2.2. Overaccumulation of Carotenoids in OC #1, OC#3, and OC#17 Line 

Carotene accumulation was determined by high-performance liquid chromatography (HPLC) using a standard curve obtained from an authentic compound. The unidentified carotenoid content eluted at 10.2, 30.2, 33.1, and 35.3 min was determined based on a standard curve for *β*-carotene. The OC#1, OC#3, and OC#17 lines accumulated L-lutein, 13Z-*β*-carotene, *α*-carotene, *β*-carotene, 9Z-*β*-carotene, and four unidentified carotenoid isomers, whereas the WT accumulated mainly unidentified carotenoids with smaller amounts of carotenoids (Figure 4). The total amount of carotenoids in OC#1, OC#3, and OC#17 lines was approximately 8.5-, 6.8- and 9.7-times higher than that of WT, respectively (Figure 4, Appendix A).

### 2.3. Analysis of Expression of Carotenoid Pathway Genes

To understand the mechanism of carotenoid accumulation in the orange calli created by *OsOr* gene editing, we determined the transcriptional levels of genes involved in carotenoid biosynthesis and catabolism by quantitative real-time PCR (Figure 5). Among the genes encoding the carotenoid metabolic pathway enzymes, the levels of *PSY2*, *PSY3*, *PDS*, *ZDS* and *β-LCY* gene transcripts were higher in the OC#1, OC#3, and OC#17 lines than in the WT. On the other hand, the transcription of *PSY1*, *CRTISO* and *ε-LCY* gene was downregulated in OC#1, OC#3, and OC#17 lines compared to that in the WT (Figure 5, Appendix A).

We also analyzed the expression levels of *CCD1*, *CCD4a*, and *CCD4b*, which encode carotenoid dioxygenase. In OC#1, OC#3, and OC#17 lines, the expression level of *CCD1* was lower than that of WT, whereas the expression levels of *CCD4a* and *CCD4b* were higher than those of WT (Figure 5, Appendix A).

### 2.4. Salt-Induces H_2_O_2_ Accumulation in OC#1, OC#3, and OC#17 Lines 

To evaluate the effects of altered OsOr expression on oxidative stress tolerance, 2-week-old transgenic calli were treated with 150 or 200 mM NaCl for 24 h. The OC#1, OC#3, and OC#17 transgenic calli were tolerant of salt-mediated oxidative stress, as determined by qualitative and quantitative analyses of H_2_O_2_ (Figure 6A,B). In addition, exposure to 200 mM NaCl resulted in a higher relative water content (RWC) in the OC#1, OC#3, and OC#17 lines than in the WT lines (Figure 6C).

## 3. Discussion

Carotenoids can be synthesized in nearly all plastids and stored at different concentrations to produce red, yellow, or orange flowers, fruits, and roots. Although it has been established that rice endosperm does not normally produce carotenoids, they have been shown to contain geranyl geranyl diphosphate (GGPP), a carotenoid precursor, and are known to express several endogenous carotenoid biosynthetic genes, including lycopene *β*-cyclase (*LCY*-*b*) [8,13]. In this regard, the endosperm of Golden rice 2, a transgenic rice variety developed by introducing a corn-derived phytoene synthase gene (*PSY*) and a soil bacterium-derived phytoene desaturase gene (*crtI*), has been shown to efficiently synthesize *β*-carotene [8]. Numerous studies have indicated that both *PSY* and orange proteins play essential roles in the accumulation of *β*-carotene [19,20,23,29,30,31]. Furthermore, it has recently been demonstrated that overexpression of the golden SNP-carrying allele of the *Or* gene results in elevated carotenoid content and orange-colored fruit or storage root; however, no significant differences in the carotenoid content of transgenic plants overexpressing the *Or* gene have been reported [20,32]. In a previous study, we successfully developed transgenic rice plants overexpressing *OsOr*-R115H #20 (leading to Arg to His substitution at position 115 on the OsOr protein) and found that there was no significant difference in the carotenoid content or calli color of these lines compared with NT plants. However, proline and chlorophyll contents of transgenic lines were significantly higher than those of non-transgenic (NT) plants under heat stress conditions [33]. Recently, Ali et al. reported that OR expression in *Arabidopsis thaliana* was upregulated by drought treatment and seedlings of OR-overexpressing (OE) lines showed improved growth performance and survival under drought stress [34]. These OE seedlings possessed lower contents of reactive oxygen species, higher activities of both superoxide dismutase and catalase, and a higher level of proline content [34].

Endo et al. achieved a high carotenoid content in orange calli by targeted mutagenesis of the *OsOr* gene using the CRISPR/Cas9 system [17]. Similar to our results, several sgRNAs were used in these experiments, but only the sgRNA of the exon 2-intron produced the orange calli phenotype. Recently, Yu et al. reported that the carotenoid content in the leaves of the *OsOr*-KO mutant line produced by CRISPR/Cas9 was similar to that of the control, whereas tillering and biomass were elevated [27].

In this study, we detected significant increases in the orange color intensity and carotenoid content, particularly that of *β*-carotene, in the OC#1, OC#3, and OC#17 lines, in which the *Or* gene was edited using the CRISPR/Cas9 system. These callus lines showed similar orange absorption spectra after several subcultures (Figure 3 and Figure 5, and Appendix A). In carrots, most orange genotypes contain *α*- and *β*-carotene, whereas yellow and red genotypes contain predominantly lutein and lycopene, respectively [35]. In tomatoes, the accumulation of carotenoids is closely associated with the expression of genes encoding biosynthetic enzymes, which is in turn influences by the size and number of plastids [36,37,38]. In addition, it has been reported that the carotenoid content in *Chrysanthemum* is determined by the expression level of carotenoid cleaved dioxygenase (*CCD*), an enzyme associated with carotenoid degradation [39]. In the present study, qRT-PCR analysis revealed differences among the three experimental rice lines with respect to the expression levels of carotenoid biosynthetic pathway genes (Figure 4), among which, we detected upregulated levels of *PSY2*, *PDS*, and *ZDS* gene transcripts in the OC#1, OC#3, and OC#17 lines. These observations indicate that carotenoid accumulation in rice calli is not directly associated with an increase in the expression of carotenoid biosynthetic genes via expression of the *Or* gene. Instead, we speculate that carotenoid synthesis in rice calli is correlated with the disruption of an undetermined domain structure in the third and fourth exons of the *Or* gene (Appendix A). This could account for the differences observed in the expression of carotenoid biosynthetic pathway genes in the OC#1, OC#3, and OC#17 lines compared with those in the WT. In addition to an increase in carotenoid content, we also found that the OC#1, OC#3, and OC#17 lines are more tolerant of salt stress than the WT (Figure 5 and Figure 6), which appears to be correlated with the increased level of carotenoids. 

We previously reported that carotenoid biosynthetic intermediates function in concert with reactive oxygen species to reduce cellular damage in plants subjected to salt stress [5,40]. Salinity is a major source of environmental stress that reduces crop productivity in arid regions worldwide [41], and evidence obtained to date indicates that enhanced tolerance to abiotic stresses, such as salt stress, is associated with plant resistance to oxidative stress [42]. Accordingly, it is conceivable that transgenic plants with enhanced resistance to different environmental stresses can be generated by manipulating genes associated with carotenoid biosynthesis.

## 4. Materials and Methods

### 4.1. Plasmid Construction and Genetic Transformation of Rice 

Target sites and single guide RNAs (sgRNAs) for the first, third, and fifth exons of *OsOr* (Os02g0651300) adjacent to a protospacer-adjacent motif (PAM) were amplified using specific primer sets (Appendix A). A CRISPR/Cas9 vector was constructed by selecting three target sites in the *OsOr* sequence using the CRISPR RGEN software (http://www.rgenome.net/ accessed on 19 November 2019) (Appendix A). A 20-nt sgRNA scaffold sequence was synthesized by Bioneer Co., Ltd. (Daejeon, Republic of Korea) and the dimer was cloned into an *Aar*I-digested OsU3:pBOsC binary vector, as described by Jung et al. [27]. The constructs obtained were transformed into rice embryogenic calli using *Agrobacterium tumefaciens* strain EHA105, as previously described [43]. Transformed calli were selected using 6 mg/L phosphinothricin and transformation was confirmed by PCR analysis, as previously reported [43]. To verify target site mutations, PCR amplicons were subjected to MiniSeq paired-end read sequencing (Illumina, San Diego, CA, USA) and analyzed using Cas-Analyzer (https://www.rgeno me.net/cas-analyzer/, accessed on 21 July 2020) [44]. All transgenic callus lines were subcultured several times and maintained in AA medium, as described previously [45] (Appendix A). 

### 4.2. qRT-PCR Analysis

Total RNA was extracted from calli using the RNeasy Plant Mini Kit (Qiagen, Hilden, Germany, www.qiagen.com (accessed on 16 December 2022)), and single-strand cDNA was synthesized with random oligonucleotides using a reverse transcription system (Takara, www.takara-bio.com (accessed on 16 December 2022)), based on a previously reported method [25]. To analyze the relative expression levels of genes involved in carotenoid biosynthesis and degradation pathways, qRT-PCR was conducted using cDNA, gene-specific primers, and SYBR Green Real-time PCR Master Mix (Toyobo, Kusatsu, Japan, http://www.toyobo.co.kr/ (accessed on 16 December 2022)), according to the manufacturer’s instructions. The sequences of gene-specific primers used for amplification are listed in Appendix A. The specificity of the amplicon was verified by dissociation curve analysis (60 to 95 °C) after 40 cycles of PCR and by agarose gel electrophoresis. The *OsActin* 1 gene was used as an internal standard and relative gene expression levels were calculated using the comparative Ct method [46].

### 4.3. Extraction of Pigments and Spectrophotometric Analysis 

Samples of rice calli (0.1 g) frozen in liquid nitrogen were homogenized using a mortar and pestle, followed by the addition of acetone (1 mL), thorough mixing, and centrifugation for 5 min at 13,000× *g* and 4 °C. Subsequently, the supernatant was collected. This procedure was repeated, and the second supernatant was combined with the first and dried. The obtained residue was dissolved in 500 μL of acetone and diluted 20 times. The absorption of the suspension was measured spectrophotometrically according to a previously reported method [47].

### 4.4. Determination of Carotenoid Content

Carotenoids of calli were extracted with 0.01% solution of butylated hydroxytoluene in acetone and analyzed using an Agilent 1260 high-performance liquid chromatography (HPLC) system (Hewlett-Packard, Waldbronn, Germany), according to the method described by Lim et al. [48]. The carotenoids were quantified using an external calibration method. The standards of *β*-carotene, *β*-cryptoxanthin, lutein, violaxanthin, and zeaxanthin were extracted using Carote Nature (Lupsingen, Switzerland). All extraction procedures were performed under low-light conditions to prevent pigment degradation and loss. Carotenoid content was calculated as µg g^−1^ dry weight of callus tissue.

### 4.5. Salt Stress 

To examine salt stress tolerance, 2-week-old calli were incubated for 24 h in liquid AA medium [45] containing 3% sucrose and 2 mg 2,4-D/l, and supplemented with 150 or 200 mM NaCl.

### 4.6. Qualitative and Quantitative Analysis of H_2_O_2_

To assess tissue oxidation, calli were placed individually in a 1 mg mL^−1^ solution of 3,3-diaminobenzidine (DAB)-HCl (pH 3.8) under continuous light, according to the methods described by Kim et al. [32]. Oxidized DAB was visualized in callus tissues as dark polymerization products generated as a result of the reaction of DAB with H_2_O_2_. The production of H_2_O_2_ was quantified by measuring the absorbance of the DAB solution at 460 nm in each callus. Oxidized DAB content was measured using a calibrated DAB standard curve.

### 4.7. Analysis of RWC

Tissue dehydration was analyzed according to the relative water content (RWC) of the transgenic calli after treatment with 150 or 200 mM NaCl. RWC was measured as follows: RWC (%) = (fresh weight − dry weight)/fresh weight [49].

### 4.8. Statistical Analyses

All data were analyzed using one-way analysis of variance (ANOVA). Subsequent multiple comparisons were performed using the least significant difference (LSD) test. According to Tukey’s HSD, statistical significance was set at *p* < 0.05.

## Figures and Tables

**Figure 1 plants-12-00056-f001:**
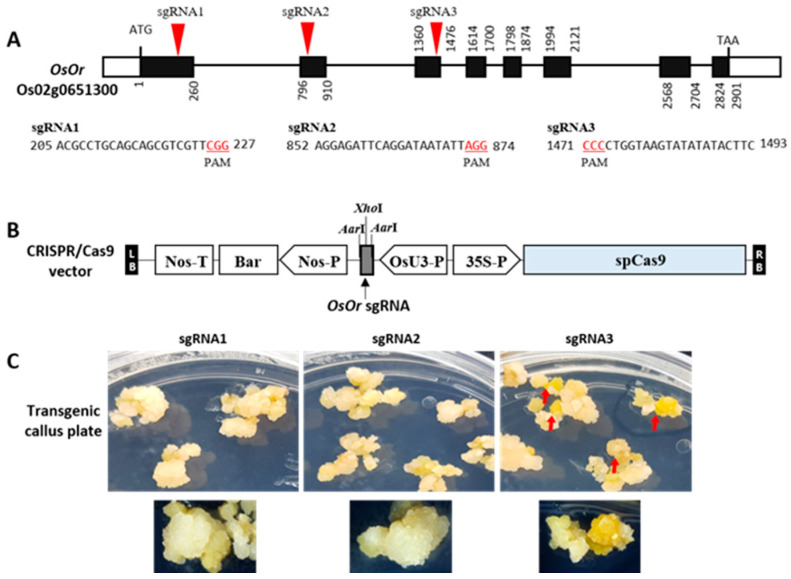
Targeted mutagenesis of *OsOr* gene in rice using CRISPR-Cas9 system. (**A**) Design of sgRNA sites for *OsOr* exons; the PAM motif (NGG) appears in red. (**B**) Schematic representation of the CRISPR-Cas9 vector construction with OsU3p:sgRNA. (**C**) Phenotype of edited callus using the CRISPR-Cas9 system.

**Figure 2 plants-12-00056-f002:**
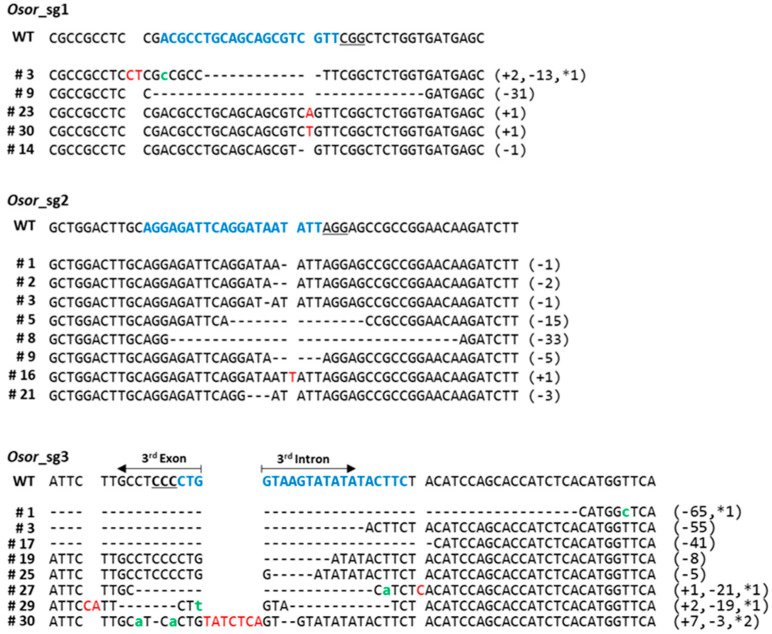
Mutation patterns in the target sequence region based on deep-sequencing analysis. The target DNA sequence of sgRNA is shown in the WT with blue text at the top of the aligned sequences. The PAM sequences are underlined. Deletions are indicated as dashes; insertions are in red; and substitutes are in green. Indel sizes are shown on the right (+, insertion; −, deletion; *, substitutes).

**Figure 3 plants-12-00056-f003:**
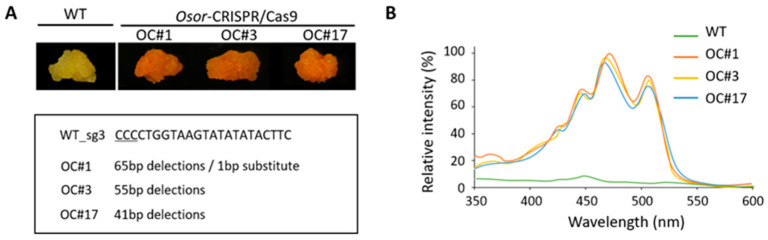
Phenotypic analysis of transgenic calli lines in rice. (**A**) Photographs of the aerial parts of 1-month-old WT, OC#1, OC#3, and OC#17 callus lines. (**B**) The absorption spectra of extracts obtained from WT (green), OC#1 (red), OC#3 (yellow), and OC#17 (blue).

**Figure 4 plants-12-00056-f004:**
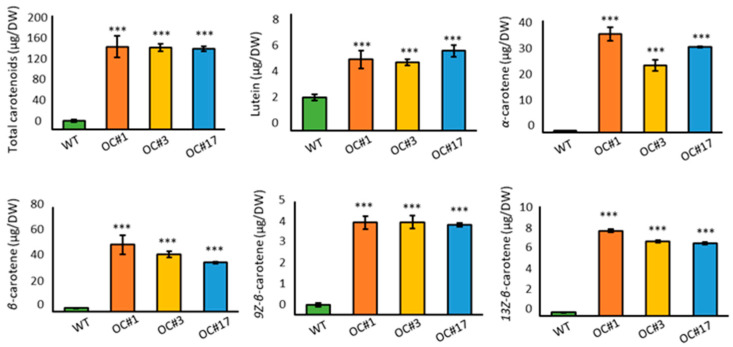
Analysis of quantitative HPLC for total carotenoids and individual carotenoid compounds in WT, OC#1, OC#3 and OC#17 calli lines. Data represent mean ± SD of three independent replicates. Asterisks indicate significant differences between OC callus line and WT callus (*** *p* < 0.001, Tukey’s HSD test).

**Figure 5 plants-12-00056-f005:**
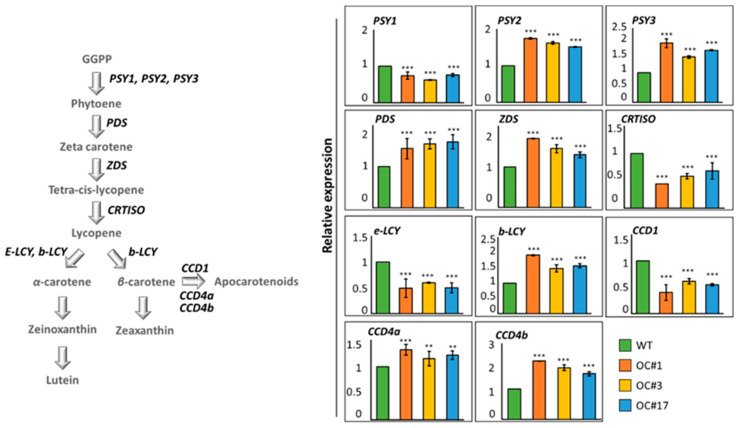
Changes in the expression of carotenoid pathway genes. The *Y*-axes of graphs indicate relative transcript amounts expressed in terms of 2^−ΔΔCT^ values determined based on qRT-PCR analyses using the *Actin 1* gene as an internal control. Data represent the means ± SD of three independent replicates. Asterisks indicate significant differences between OC callus line and WT callus (** *p* < 0.01, *** *p* < 0.001, Tukey’s HSD test).

**Figure 6 plants-12-00056-f006:**
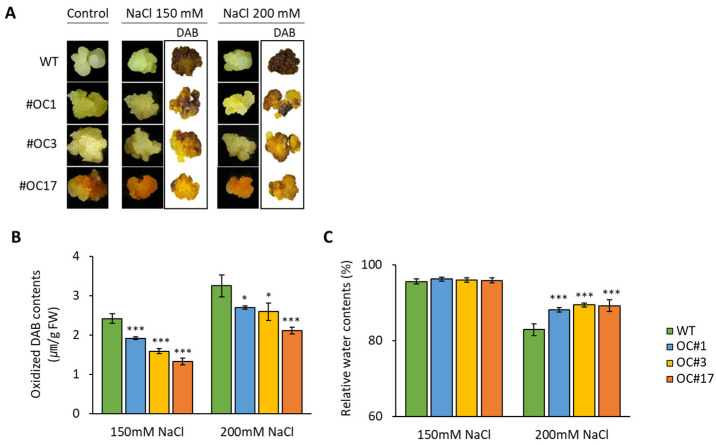
Effect of salt-mediated oxidative stress on transgenic calli lines after NaCl treatment for 24 h. (**A**) Phenotypical analysis of visible damage in WT, OC#1, OC#3, and OC#17 callus lines after stress treatment with 150 mM and 200mM NaCl by DAB staining. (**B**) Oxidized DAB contents. (**C**) Relative water content (RWC) of transgenic calli lines. Data are expressed as the mean ± SD values of three independent replicates SD. Asterisks indicate significant differences between OC callus line and WT callus (* *p* < 0.05, *** *p* < 0.001, Tukey’s HSD test).

## Data Availability

Not applicable.

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
