# Peer review of "Biochemical Characterization of Orange-Colored Rice Calli Induced by Target Mutagenesis of OsOr Gene"

_plants, 2022, doi:10.3390/plants12010056_

Round 1

Reviewer 1 Report

Overall, the methodology used in this study is relatively consistent, with common controls (3 mutant lines, visual phenotype, metabolic quantifications related to the phenotype and even qPCR analysis of carotenoid pathway).  I have to say I was a bit surprise by the use of rice calli, even if it was specifically explained in the title of the manuscript. Obviously, when the authors are introducing the literature, many attempts were done in other plants but without using calli only. Thus, we expected that the study was using “real plants”, to provide a good potential application for next-generation breeding approaches.  Then I found this article published in 2021 (https://doi.org/10.1016/j.plantsci.2021.110962 “ Overexpression of the rice ORANGE gene OsOR negatively regulates carotenoid accumulation, leads to higher tiller numbers and decreases stress tolerance in Nipponbare rice”). I think this published work should be clearly introduce in the introduction and should stimulate the research framework proposed by the authors, i.e. to perform mutant lines to get orange calli, in opposition to the previous results.

Besides this, I have the following comments:

-        The first paragraph introduces many figures that are cited few pages after. This make the results a bit difficult to follow. Can you cut the text with figures more accurately to help the reader please ?

-        The analysis of carotenoid content must be a bit more detailed. Some transgenic lines statistically accumulated different levels of some metabolic intermediates. Please mention this and perhaps add a brief explanation with respect to the metabolic pathway. I think it could be interesting to add a summary graph showing the carotenoid distribution in % for Wt and mutant lines. This will help to disentangle whether the small statistical differences between the transgenic lines have an impact on the final distribution or not.

-        The analysis of carotenoid gene expression must be a bit more detailed. What about the intensity of these gene regulations ? TO whom it may concern, a factor 2 for regulation of gene expression is relatively weak. However, look at the strong differences of carotenoid accumulation. I think this can deserve mode text. In addition, the PSY genes were not regulated the same way. Please mention this and perhaps add a brief explanation if necessary.

-        Figure 4 and Figure 5: Please add the bars for y-scales in front of each number.

-        Figure 5: I do not understand the relative expression values. Usually, the Wt gene expression value is set to 1 and other groups are reported relatively to the Wt. Why are they different values for Wt. Can you please fix this important issue ?

-        Figure 5 : Please add the gene names to the metabolic pathway to help the readers to understand qPCR results.

-        qPCR were performed with only one reference gene. This is extremely controversial. The use of 2 reference genes is a pre-requisite because the variation of one reference gene is never enough stable for different lines/conditions.

-        How the qPCR primers were selected. What were there efficiencies ? the authors used the 2deltadeltaCt methodology, which assume that all primers efficiencies are equal to 2. To whom it may concern, this is never absolutely true. Please explain which cut-off values you used to select them. How many qPCR cycles were performed ? did you do a melting curve to confirm the specific amplification of only one PCR product ?  Please add more details…

-        L287: what is “dim ligh”. Please change this term to something more readable.

-        L259: the reference (45] seems a bit confusing compared to the reference [49] L291. I assume that Sucrose was present in all media used here ? Can you please use only 1 reference for media composition and explain it in details in the L259 ?

L236: the “.” to remove

Author Response

Response to Comments

We appreciate the comments that the reviewers have given in our manuscript and the constructive criticism they have given. We have carefully reviewed the comments and have revised the manuscript accordingly. We believe that these changes have clearly improved our manuscript.  

Reviewer 1

  1. The article published in 2021 (https://doi.org/10.1016/j.plantsci.2021.110962 “Overexpression of the rice ORANGE gene OsOR negatively regulates carotenoid accumulation, leads to higher tiller numbers and decreases stress tolerance in Nipponbare rice”). I think this published work should be clearly introduce in the introduction

>> Thank you for kindly reviewing the author. The authors inserted the paper presented by the Chinese group into line 85-87 in detail as follows: In addition, Yu et al. reported that overexpression of the rice orange gene OsOr negatively regulated carotenoid accumulation, leading to higher tiller numbers and decreased stress tolerance in Nipponbare rice.

  1. The first paragraph introduces many figures that are cited few pages after. This make the results a bit difficult to follow. Can you cut the text with figures more accurately to help the reader please?

>> Yes, the figures are distributed and arranged according to the pages of the manuscript.

  1. Authors please write results for Figure 1C.

>> Thank you for kindly reviewing the author. The authors inserted the description of Figure 1C in lines 107-109 as follows: Among them, the orange color phenotype was observed in the transgenic callus obtained from the sgRNA3 construct (Figure 1 C).

  1. The analysis of carotenoid content must be a bit more detailed. Some transgenic lines statistically accumulated different levels of some metabolic intermediates. Please mention this and perhaps add a brief explanation with respect to the metabolic pathway. I think it could be interesting to add a summary graph showing the carotenoid distribution in % for Wt and mutant lines. This will help to disentangle whether the small statistical differences between the transgenic lines have an impact on the final distribution or not.

>> The authors have previously written many papers on carotenoid analysis. Therefore, carotenoid content analysis is routinely conducted. However, the authors committed mistakes in statistical analysis. Therefore, the statistical analysis was performed again and corrected in Figure 4.

>> A brief description of the metabolic pathway was also provided in lines 218-227 of the discussion.

>> The authors also provided the distribution of carotenoids for WT and mutant lines in Supplementary Table 4 for numerical comparison.

  1. The analysis of carotenoid gene expression must be a bit more detailed. What about the intensity of these gene regulations? TO whom it may concern, a factor 2 for regulation of gene expression is relatively weak. However, look at the strong differences of carotenoid accumulation. I think this can deserve mode text. In addition, the PSY genes were not regulated the same way. Please mention this and perhaps add a brief explanation if necessary.

>> Thank you for kindly reviewing the author. Reflecting the reviewer's opinion, the authors marked lines 185-187 as follows: On the other hand, the transcription of PSY1, CRTISO and ε-LCY gene were downregulated in OC#1, OC#3, and OC#17 lines compared to that in the WT (Figure 5, Supplementary Table 6).

>> In addition, it was pointed out that the gene expression level was weak compared to the difference in carotenoid content, because the Or gene was not the main gene for carotenoid biosynthesis. The Or gene is known as a cysteine-rich DnaJ protein that localizes to plastids and regulates carotenoid accumulation.

  1. Figure 4 and Figure 5: Please add the bars for y-scales in front of each number.

>> Thank you for kindly reviewing the author. Authors added comments made by reviewers.

  1. Figure 5: I do not understand the relative expression values. Usually, the Wt gene expression value is set to 1 and other groups are reported relatively to the Wt. Why are they different values for Wt. Can you please fix this important issue?

>> The authors made a mistake in the graphical representation of qRT-PCR results. Therefore, according to the reviewer's point, the expression of WT was re-marked as 1.

  1. Figure 5: Please add the gene names to the metabolic pathway to help the readers to understand qPCR results.

>> Thank you for kindly reviewing the author. The authors marked the gene names on metabolic pathways.

  1. qPCR was performed with only one reference gene. This is extremely controversial. The use of 2 reference genes is a pre-requisite because the variation of one reference gene is never enough stable for different lines/conditions.

>> I totally agree with what the reviewer pointed out. However, in many reported papers, the relative expression level is calculated using the actin gene (structural gene) as a reference gene to calculate the expression level. Therefore, the authors calculated gene expression values using actin as a reference gene.

  1. How the qPCR primers were selected. What were there efficiencies? the authors used the 2deltadeltaCt methodology, which assume that all primers efficiencies are equal to 2. To whom it may concern, this is never absolutely true. Please explain which cut-off values you used to select them. How many qPCR cycles were performed? did you do a melting curve to confirm the specific amplification of only one PCR product? Please add more details.

>> The design of qPCR primers was performed in Primer 3 (https://primer3.org), and the specificity of the primers was indicated in lines 315-316 in Materials and Methods as follows: The specificity of the amplicon was verified by dissociation curve analysis (60 to 95 °C) after 40 cycles of PCR and by agarose gel electrophoresis.

>> Until now, in many papers, the calculation of qRT-PCR analysis was usually quantified with values of 2-ΔΔCT. Therefore, in this paper, it was calculated by converting to the values of 2-ΔΔCT.

>> Upon completion of qRT-PCR analysis on the PCR instrument, expression levels are displayed based on melting curve values. Therefore, the 2-ΔΔCT value is calculated from these values and the expression level of the reference gene. Therefore, melting curves were confirmed in all experiments.

  1. L287: what is “dim ligh”. Please change this term to something more readable.

>> Thank you for kindly reviewing the author. The authors changed dim light to low-light.

  1. L259: the reference (45] seems a bit confusing compared to the reference [49] L291. I assume that Sucrose was present in all media used here? Can you please use only 1 reference for media L259:

>> Thank you, author for your kind review. The authors rewritten using one reference for AA medium.

Reviewer 2 Report

The DnaJ-like protein ORANGE has been reported for 16 years, and has also been widely studied for its functions in triggering carotenoid accumulation. In this study, the authors reported their results of inducing carotenoid accumulation and salt resistance in rice callus by the mutagenesis induced by the CRISPR/Cas9 technology. Although previous reports showed that either a transposon insertion (the original cauliflower mutation), or a different spliced version, or the golden SNP of the orange gene could result in similar phenotypes, this study might show a novel way of how OR functions, which is the novelty of this work. However, as the novelty relies on the mutation, the authors should provide detailed information about the mutations in their transgenic lines. Since they have sequenced the region, it should be easy to provide an alignment of the mutant versions and the WT ORANGE protein sequence, so that which mutant version(s) of OR is responsible for carotenoid accumulation and drought resistance will be clear. This is my major concern which has to be resolved.

My other minor concerns include:

1. The references were not listed correctly. For example, reference 18 was cited in line 61 for a transgenic tomato work, but this reference is on Chlamydomonas. In Line 63, reference 19 should be 20. This needs to be carefully checked.

2. The writing needs to be carefully checked. Some sentences are incomplete. Calli is the plural of callus, and maybe cannot be used as an adjective. In line 38, "harvest" should be "harvest".

3. In line 41, I suggest using "quenching" or some similar words to substitute "reacting".

4. Figure 1. Panel 1C does not provide much information. Does it mean one line was selected from each sgRNA-transformed material? What are their mutations? Are there other lines having the same mutation but did not show similar phenotypes?

5. Figure 2 can be moved to supplemental files. Sequence alignment and photographs of transgenic calli can be put here.

6. Figure 3 does not provide any information. HPLC elution profile will be better.

7. Figure 4, numbers in the Y-axis were not aligned to where they should be. And the Y-axis needs ticks!

8. Figure 5. Please label each of the genes quantified into the pathway.

9. Figure 6. Non-NaCl-treated control is missed.

10. Discussion. The first paragraph just repeated introduction.

11. A recent work (Ali et al. Int J Mol Sci . 2022. 23:3907) showing the OR regulates proline biosynthesis should be cited and discussed.

Author Response

Response to Comments

We appreciate the comments that the reviewers have given in our manuscript and the constructive criticism they have given. We have carefully reviewed the comments and have revised the manuscript accordingly. We believe that these changes have clearly improved our manuscript.  

Reviewer 2

The authors should provide detailed information about the mutations in their transgenic lines.

>> Thank you for your kind review. The authors inserted to line 133-134 to the mutation information as indicated by the reviewers as follows: Among sgRNA3-derived transgenic calli, three independent orange calli ( -65*1/-65*1, -55/-55, -41/-41) with large deletions were selected.

My other minor concerns include:

  1. The references were not listed correctly. For example, reference 18 was cited in line 61 for a transgenic tomato work, but this reference is on Chlamydomonas. In Line 63, reference 19 should be 20. This needs to be carefully checked.

>> Thank you for your kind review. References were reorganized according to the reviewer's comments.

  1. The writing needs to be carefully checked. Some sentences are incomplete. Calli is the plural of callus, and maybe cannot be used as an adjective. In line 38, "harvest" should be "harvest".

>> Thank you for your kind review. As pointed out by the reviewers, the authors performed English proofreading on Editage (direct@editage.com) to correct and supplement the manuscript.

  1. In line 41, I suggest using "quenching" or some similar words to substitute "reacting".

>> The author has replaced it with "interacting".

  1. Figure 1. Panel 1C does not provide much information. Does it mean one line was selected from each sgRNA-transformed material? What are their mutations? Are there other lines having the same mutation but did not show similar phenotypes?

>> Thank you for kindly reviewing the author. The authors inserted the description of Figure 1C in lines 107-109 as follows: Among them, the orange color phenotype was observed in the transgenic callus obtained from the sgRNA3 construct (Figure 1 C).

  1. Figure 2 can be moved to supplemental files. Sequence alignment and photographs of transgenic calli can be put here.

>> The authors show the variant types of sgRNA1, sgRNA 2, and sgRNA3 in Figure 2.

  1. Figure 3 does not provide any information. HPLC elution profile will be better.

>> Figure 3 was supplemented with the mutation types of OC#1, OC#3 and OC#17 lines.

  1. Figure 4, numbers in the Y-axis were not aligned to where they should be. And the Y-axis needs ticks!

>> Thank you for kindly reviewing the author. Edited according to the reviewer's comments.

  1. Figure 5. Please label each of the genes quantified into the pathway.

>> Thank you for kindly reviewing the author. Edited according to the reviewer's comments.

  1. Figure 6. Non-NaCl-treated control is missed.

>> Thank you for kindly reviewing the author. Edited with additions based on reviewer comments.

  1. The first paragraph just repeated introduction.

>> Removed based on reviewer comments.

  1. A recent work (Ali et al. Int J Mol Sci . 2022. 23:3907) showing the OR regulates proline biosynthesis should be cited and discussed.

>> Thank you for kindly reviewing the author. The authors explained in lane 237-241 by citing recent papers on Or gene and proline as follows: Recently, Ali et al. report that OR expression in Arabidopsis thaliana was up-regulated by drought treatment and seedlings of OR-overexpressing (OE) lines showed improved growth performance and survival under drought stress. These OE seedlings possessed lower contents of reactive oxygen species, higher activities of both superoxide dismutase and catalase, and a higher level of proline content [34].

Round 2

Reviewer 2 Report

The authors have answered my questions and did revisions accordingly. I don;t have further comments, and would like to recommend an acceptance of this manuscript for publication.